# The Influence of Gut Dysbiosis in the Pathogenesis and Management of Ischemic Stroke

**DOI:** 10.3390/cells11071239

**Published:** 2022-04-06

**Authors:** Saravana Babu Chidambaram, Annan Gopinath Rathipriya, Arehally M. Mahalakshmi, Sonali Sharma, Tousif Ahmed Hediyal, Bipul Ray, Tuladhar Sunanda, Wiramon Rungratanawanich, Rajpal Singh Kashyap, M. Walid Qoronfleh, Musthafa Mohamed Essa, Byoung-Joon Song, Tanya M. Monaghan

**Affiliations:** 1Department of Pharmacology, JSS College of Pharmacy, JSS Academy of Higher Education & Research, Mysuru 570015, Karnataka, India; ammahalakshmi@jssuni.edu.in (A.M.M.); sonalisharma578@gmail.com (S.S.); tousif.a.h7@gmail.com (T.A.H.); bray365@gmail.com (B.R.); tuladharsunanda4@gmail.com (T.S.); 2Centre for Experimental Pharmacology and Toxicology, Central Animal Facility, JSS Academy of Higher Education & Research, Mysuru 570015, Karnataka, India; 3Food and Brain Research Foundation, Chennai 600094, Tamil Nadu, India; agrathipriya@gmail.com; 4Section of Molecular Pharmacology and Toxicology, Laboratory of Membrane Biochemistry and Biophysics, National Institute on Alcohol Abuse and Alcoholism, National Institutes of Health, Rockville, MD 20892, USA; wiramon.rungratanawanich@nih.gov; 5Research Centre, Dr G. M. Taori Central India Institute of Medical Sciences (CIIMS), Nagpur 440 010, Maharashtra, India; raj_ciims@rediffmail.com; 6Q3CG Research Institute (QRI), Research & Policy Division, 7227 Rachel Drive, Ypsilanti, MI 48917, USA; walidq@yahoo.com; 721 HealthStreet, Consulting Services, 1 Christian Fields, London SW16 3JY, UK; 8Department of Food Science and Nutrition, CAMS, Sultan Qaboos University, Muscat 123, Oman; 9Aging and Dementia Research Group, Sultan Qaboos University, Muscat 123, Oman; 10National Institute for Health Research Nottingham Biomedical Research Centre, University of Nottingham, Nottingham NG7 2UH, UK; 11Nottingham Digestive Diseases Centre, School of Medicine, University of Nottingham, Nottingham NG7 2UH, UK

**Keywords:** cerebral stroke, gut microbiota, gut dysbiosis, gut-derived metabolites, gut leakiness, gut–brain axis, gut immune cells

## Abstract

Recent research on the gut microbiome has revealed the influence of gut microbiota (GM) on ischemic stroke pathogenesis and treatment outcomes. Alterations in the diversity, abundance, and functions of the gut microbiome, termed gut dysbiosis, results in dysregulated gut–brain signaling, which induces intestinal barrier changes, endotoxemia, systemic inflammation, and infection, affecting post-stroke outcomes. Gut–brain interactions are bidirectional, and the signals from the gut to the brain are mediated by microbially derived metabolites, such as trimethylamine N-oxide (TMAO) and short-chain fatty acids (SCFAs); bacterial components, such as lipopolysaccharide (LPS); immune cells, such as T helper cells; and bacterial translocation via hormonal, immune, and neural pathways. Ischemic stroke affects gut microbial composition via neural and hypothalamic–pituitary–adrenal (HPA) pathways, which can contribute to post-stroke outcomes. Experimental and clinical studies have demonstrated that the restoration of the gut microbiome usually improves stroke treatment outcomes by regulating metabolic, immune, and inflammatory responses via the gut–brain axis (GBA). Therefore, restoring healthy microbial ecology in the gut may be a key therapeutic target for the effective management and treatment of ischemic stroke.

## 1. Introduction

Stroke is the second and third leading causative factor for death and disability, respectively [1], and imposes huge socioeconomic and medical problems [2]. Ischemic stroke causes a major burden on society, with an annual rate of 24.9 million cases worldwide [3]. Clinically, stroke is defined as a brain tissue injury caused by a lack of blood supply to a specific region(s), which results in permanent neuronal impairment or death [3,4,5]. Cerebral ischemic stroke, which accounts for 85% of all strokes, is caused by vascular occlusion or stenosis of an artery, whereas haemorrhagic stroke, which accounts for 15% of strokes, is triggered by vascular rupture, resulting in intraparenchymal and/or subarachnoid hemorrhage [1].

Although intravenous thrombolysis, endovascular thrombectomy, and stroke therapeutic agents can improve physical, motor-behavioural, and mental deficits in some patients, the prognosis for most post-stroke patients is very poor [6,7]. Several studies found that 90% of stroke cases are correlated with behavioral factors, such as unhealthy Western-style high-fat diets containing omega-6 fatty acids, smoking, alcohol, and low physical activity. Stroke is also commonly associated with metabolic causes, including insulin resistance, obesity, hypertension, and type 2 diabetes mellitus (T2DM), all of which have major influences on the composition and abundance of gut microbiota [8,9,10]. Hence, identification of the potential risk factors affecting stroke prognosis and underlying pathogenic mechanisms is critically important for improved management and treatment of stroke [11,12].

Recent studies have demonstrated the existence of bidirectional communications between the gut and brain through gut microbiota. The intimate bidirectional interactions between the gut and brain occur through the neural (central, autonomic, and enteric nervous systems), hormonal (endocrine system), and immunological (innate and acquired immune system) routes [13,14], which are commonly termed the microbiota–gut–brain axis or the gut–brain axis (GBA) [15]. The gut microbiota and microbially derived metabolites play a crucial role in brain functions by regulating GBA signaling [16,17]. Recent evidence also indicates that the GBA plays a central role in regulating immune function(s) in post-stroke conditions [18].

Interestingly, several studies have demonstrated the direct influence of gut dysbiosis on the development of clinical risk factors for stroke (e.g., Western-style high-fat diets, hypertension, T2DM, obesity, insulin resistance, metabolic syndromes, dyslipidemia, atherosclerosis, aging, vascular dysfunction, inflammation, gut leakiness, and alcohol excess [19,20,21,22,23]. Henceforth, the gut microbiome has recently been emerged as a new important “organ” of the human organism [24,25], as it can influence the early stages of neural, immune, and metabolic abnormalities and pathologies [16,26,27,28]. Recent research has demonstrated that an altered gut microbiome influences post-stroke outcomes through multiple factors, including local and systemic inflammation, gut leakiness, endotoxemia, bacterial components, metabolites, and immune and neural systems. Hence, the pathogenic association of stroke-induced gut dysbiosis with poor treatment outcomes has been actively explored in recent years in the hope of leading to more effective therapies. 

Stroke causes gut dysmotility and increases gut-barrier permeability and translocation of the microbes and microbially derived products, such as lipopolysaccharide (LPS) and trimethyl amine-N-oxide (TMAO), into the bloodstream. These changes accelerate systemic inflammation and the worsening of symptoms, often contributing to poor prognosis [29]. Notably, gut dysbiosis exerts chronic inflammatory responses both peripherally and centrally that accelerate stroke pathology [30]. Collectively, gut microbial dysbiosis is considered a major risk factor that correlates positively with poor post-stroke outcomes. In this review, we summarize preclinical and clinical evidence that demonstrates the pathogenic influence of gut dysbiosis in cerebral ischemic stroke. We also explore the potential of the gut microbiota as a novel therapeutic target for ischemic stroke.

## 2. Gut Microbiota 

The human gastrointestinal (GI) tract constitutes trillions of microorganisms, such as viruses, bacteria, fungi, and protozoa [31,32]. Microbes residing in the GI tract are commonly termed gut microbiota and, combined with their functional characteristics and genetic materials, are termed the gut microbiome [16,33,34]. Gut microbes are known to maintain body homeostasis by regulating digestive, metabolic, immune, and neurological functions [8,35,36] via the highly interconnected complex physiological pathways called the gut–brain axis or GBA [37,38]. Gut microbes maintain the integrity of the intestinal epithelial barrier and stimulate intestinal cell regeneration and production of mucin and other metabolites, including bile acids, ethanol, acetaldehyde, acetate, and other SCFAs [39]. The GI tract is considered a major immune organ, containing the largest pool of immune cells, constituting more than 70% of the entire immune system. The majority of gut bacteria belong to either Bacteroidetes or Firmicutes phyla (about 51% or 48%, respectively), whereas Actinobacteria (including the *Bifidobacteria* genera), Spirochaetes, Proteobacteria, Cyanobacteria, Leptosphaeria, Fusobacteria, and Verrucomicrobia phyla are found to be at relatively lower abundance [35,40]. In addition, the gut microbiota is a central regulator of T-cell homeostasis and plays a crucial role in the maturation of the immune system [41,42,43]. The pathological alterations in the composition (or diversity) and abundance of gut microbes, leading to altered gut-immune and neuroimmune status, is termed gut dysbiosis or gut microbial dysbiosis [44]. Gut dysbiosis, frequently associated with elevated intestinal-barrier dysfunction and local inflammation [45,46,47], usually contributes to disrupted GBA signaling, which results in pathophysiological consequences [48]. Increased translocation of bacteria and their toxic products into systemic circulation predisposes to a vast array of GI, metabolic, endotoxemia, cardiovascular, and neurological disorders [46,48,49].

### 2.1. Microbially Derived Molecules

Gut microbes produce essential bioactive metabolites that play a crucial role in the maintenance of homeostasis, immune maturation, mucosal integrity, and host energy metabolism [39,50]. Short-chain fatty acids (SCFAs), such as butyrate, acetate, and propionate, are the main metabolites found in the colon produced by anaerobic fermentation of dietary fibres and resistant starch [27,51]. These SCFAs serve as an important energy source and also possess immunomodulatory and neuroactive properties [14,50]. The SCFAs regulate host cell physiological functions via a variety of mechanisms, including histone acetylation-related epigenetic alteration, cell proliferation, and G-protein-coupled receptor activation. Additionally, SCFAs maintain glucose metabolism [52,53], inflammation [54], and blood pressure [55], and regulate the maintenance of the blood–brain barrier (BBB) [56] and the physiology of microglia [57]. Reduced abundance of SCFA-producing bacteria is observed in metabolic diseases, as well as cardiovascular and neuropsychiatric disease models, including stroke, hypertension, insulin resistance, obesity, and T2DM [58,59,60]. These intestinal microbes are involved in the production of vitamins B and K, as well as the absorption and metabolization of essential substances, such as bile acids, sterols, and drugs. Gut bacteria are also known to synthesize and/or stimulate a number of neurotransmitters, such as acetylcholine; γ-aminobutyric acid (GABA); serotonin (5-HT); melatonin and its precursor, N-acetyl serotonin; glutamate; dopamine; and noradrenaline [61,62,63], and to modulate activation of the immune system [43,57]. In addition, SCFAs interact with G-protein-coupled receptors on enteroendocrine cells (specialized intestinal cells) and trigger the secretion of gut hormones, such as glucagon-like peptide-1 (GLP-1), cholecystokinin (CCK), and peptide YY (PYY), either indirectly (via systemic circulation) or directly (via the vagus nerve) through the GBA. Gut bacterial metabolites, such as SCFAs, nitrites, TMAO, indoles, and hydrogen sulphide, are found to have profound effects on the cardiovascular and cerebrovascular systems. For example, SCFAs and hydrogen sulphide possess vasorelaxant properties, and the abundances of bacterial species, such as Gram-positive Firmicutes [64], which produce these beneficial compounds, are found to be decreased in hypertensive animal models and human studies compared to healthy counterparts [15,65]. In contrast, higher levels of TMAO, a metabolite that can increase endothelial reactive oxygen species (ROS) production and impair endothelially mediated vasodilation [66], are found to be positively correlated with adverse cardiovascular events, including ischemic stroke [67,68] in patients with atherosclerosis [69], thrombosis due to enhanced platelet hyperactivity [70], and atrial fibrillation [71]. Thus, TMAO is considered a novel predictor of stroke. LPS, a component of the outer membrane of Gram-negative bacteria, often called an endotoxin, can enter blood circulation when the intestinal-barrier integrity is disrupted by bacterial translocation and gut dysbiosis, leading to systemic inflammation [72,73] (Table 1). 

### 2.2. Gut–Brain Axis

The gut–brain axis (GBA) refers to the bidirectional link between the GI tract and the central nervous system (CNS) [16,79]. Recent studies in the field of neuroscience and neuroimmunology [79,85] have revealed that the functional crosstalk between the gut microbiota and brain through GBA signaling is mediated by complex and multiple pathways. The top-to-bottom pathways from the brain to the gut influence sensory, motor, and secretory modalities of the GI tract [86,87], whereas the bottom-to-top signals from the gut to the brain impact cognitive and neurobehavioral functions [8,88]. The precise pathways involved in the GBA remain to be fully determined. However, recent studies have shown that the GBA includes both neuronal and non-neuronal pathways through which the gut microbiota affect brain functions and vice versa [17,89,90].

#### 2.2.1. Brain-to-Gut Signaling or Top-Down Pathway

In the top-down pathway, the brain influences the gut wall via both direct and indirect pathways. (i) Direct pathways include (a) the extrinsic parasympathetic and sympathetic branches of the autonomic nervous system and (b) the endocrine system (HPA axis signals from brain to gut are altered, particularly in response to a variety of stressful stimuli) [91,92,93]. Furthermore, activation of the afferent fibres activates the HPA axis, which in turn stimulates the brain areas, such as hypothalamic neurons that regulate pituitary secretions and the nucleus tractus solitarii, with its downstream projections [39,94]. (ii) Indirect pathways include (a) the intrinsic branches of the enteric nervous system (a highly developed system of neuronal connections located in the submucosa and myenteric plexus of the gut wall) [56,95] and (b) the immune system—the immune cells of the central nervous system (such as microglia and astrocytes). These neural connections regulate nutrient absorption; metabolism; peristaltic movements; permeability changes; mucus secretion from goblet cells; microbiome composition and abundance; secretion of neurotransmitters or neuropeptides, such as dopamine, serotonin (5-HT), norepinephrine, and epinephrine; stress hormones (release of CRH, ACTH, and cortisol); and resident immune cell activation by secreting various cytokines and chemokines [78].

#### 2.2.2. Gut-to-Brain Signaling or Bottom-up Pathway

Bottom-up signals are carried by afferent vagal branches to the brain [96]. Primarily, the vagus nerve (made up of 80% afferent and 20% efferent fibres) serves as a major neural bidirectional communication route between the gut and brain (both directly and indirectly) [97,98]. The hepatic and celiac branches of the vagus nerve are stimulated by microbially derived substances (e.g., LPS, an endotoxin from the outer membrane of Gram-negative bacteria; nitric oxide; indoles; and bile acids), metabolites (e.g., SCFAs and TMAO), and hormones [99,100] released from enteroendocrine cells of the gut epithelial layer [27,101,102]. These bacterial metabolites, which are secreted into the systemic circulation, travel to the brain and modulate the function of neurons, astrocytes, microglia, and the BBB [57,103,104]. Bottom-up signaling regulates gut-barrier integrity via tight junctions, such as occludin; release of neurotransmitters, such as 5-HT, GABA, and catecholamines; as well as gut hormones, such as CCK, PYY, and GLP-1. It also regulates immune and inflammatory responses by controlling the innate and adaptive immune cell responses.

Altogether, these bidirectional interactions highlight the crucial role of GBA in regulating host health and its involvement in association with the plethora of diseases, ranging from cardiovascular complications to neurologic illnesses [79] and neuropsychiatric disorders [105]. As a result, dysregulated GBA signaling, either due to altered composition of the gut bacteria or metabolites, influences stroke pathogenesis and impairs post-stroke outcomes. Hence, the gut microbiota in the GBA represents an attractive target for the development of novel treatment approaches in the form of special nutritional and other interventions, such as prebiotics, probiotics, faecal microbial transplantation (FMT), and synbiotics [30,79].

## 3. Gut Microbial Dysbiosis and Cerebral Stroke

### 3.1. The Role of Gut Dysbiosis in Stroke

The prominent role of gut dysbiosis in stroke can be explained by answering three important questions.

(a)How does stroke alter the GM?

In stroke-induced gut dysbiosis, the ischemic brain dysregulates GBA signaling either via the neural or HPA axis pathways. Metabolic endotoxemia occurs to elevated levels of plasma LPS, which contributes to increased intestinal permeability [106]. Metabolic endotoxemia activates the innate immune cells, contributing to chronic systemic inflammation [107,108] with increased migration of immune cells into the brain, resulting in the breakdown of the BBB and neuroinflammation [44,78]. Experimental stroke models in rodents have clearly indicated decreased diversity of the intestinal microbiome following stroke [109]. Clinical trials in stroke patients [110,111,112,113] have shown gut dysbiosis with alterations in the Firmicutes-to-Bacteroidetes ratio, increased abundance of opportunistic pathogens (*Megasphaera*, *Enterobacter*, and *Desulfovibrio*), and decreased abundance of beneficial SCFA-producing bacteria (*Blautia*, *Roseburia, Anaerostipes, Bacteroides*, *Lachnospiraceae*, and *Faecalibacterium*). A clinical study investigating the microbiome of patients after cerebral stroke showed a reduced abundance of Roseburia, Bacteroides, and *Faecalibacterium prausnitzii* and an increased abundance of *Enterobacteriaceae*, *Bifidobacteriaceae*, and *Clostridium difficile* in intestinal samples when compared to healthy subjects, intensive care patients, and patients with active ulcerative colitis or irritable bowel syndrome [114]. Symptomatic patients with transient ischemic attack or mild ischemic stroke exhibited altered intestinal microbiomes compared to patients with asymptomatic atherosclerosis and without atherosclerotic changes [29,114,115]. 

(b)How doe GM influence the stroke outcome/prognosis?

Approximately 50% of stroke patients report GI symptoms, supporting the presence of gut dysbiosis [116]. Ischemic stroke directs aberrated signals to the intestine via GBA, which results in defective intestinal-barrier integrity, reduced mucus secretion, and translocation of intestinal bacteria into circulation and the extraintestinal organs [78,109]. Gut dysbiosis affects the local immune cells in the intestine and brain. In the early phase of stroke, microglial activation is followed by infiltration of peripheral immune cells, especially monocytes, as well as T- and B-lymphocytes [117,118]. In mouse stroke models, gut dysbiosis intensifies the ingress of Th17- and IL17-secreting γδ T-cells (γδ T-cells) into the CNS from the intestine, leading to chronic systemic and neuroinflammation. Higher numbers of proinflammatory lymphocyte populations correlate negatively with stroke outcome, which is reflected as larger infarct size, brain edema, and neurological deficits [44,78]. 

(c)How does GM contribute to stroke pathology or pathogenesis of risk factors for stroke development?

Reports from rodent and human clinical studies indicate that gut dysbiosis influences several pathways implicated in the development of risk factors associated with cerebral stroke, such as hypertension and metabolic diseases like obesity, T2DM, atherosclerosis, and vascular dysfunctions [20,21,22]. Further metabolic disorders are accompanied by increased plasma endotoxin LPS levels, suggesting increased gut leakiness. For instance, an altered ratio of the Bacteroidetes to Firmicutes, along with enhanced capacity to gain energy from the diet, are found in genetically obese mice and human volunteers. Germ-free (GF) mice receiving FMT from obese mice show a marked increase in total body fat compared to GF mice receiving FMT from lean mice, suggesting an important role of gut microbiota in regulating metabolic states [119]. 

Altogether, these results explain the pathogenic role of gut dysbiosis in the onset, progression, and outcomes of stroke; therefore, potential therapeutic approaches targeting gut dysbiosis can be considered in the treatment and management of stroke. 

### 3.2. Pathogenic Immune Signals in Stroke-Induced Gut Dysbiosis 

The key inflammatory and immune cells involved in stroke and gut dysbiosis are: (a)Macrophages and monocytes;(b)T lymphocytes, such as CD4^+^ T helper (Th) cell subsets (Th1-, Th17-, and IL17-secreting γδ T-cells), CD8^+^ T cells, Treg cells, and natural killer T-cells;(c)B lymphocytes;(d)Microglia;(e)Astrocytes;(f)Dendritic cells;(g)Neutrophils;(h)Mast cells.

#### 3.2.1. Innate Immune Signaling

The pathophysiology of stroke-induced gut dysbiosis includes a series of inflammatory responses induced by the activation of gut immune cells, increased gut permeability with elevated endotoxin levels, and their trafficking pathways to the brain. After stroke, the innate immune response is produced by innate immune cells, such as neutrophils, macrophages, microglia, mast cells, lymphocytes (γδ T-cells), and natural killer T-cells, followed by the activation of the adaptive immune response, which is mainly mediated by T and B lymphocytes [78]. The pathological cascade following stroke begins with the release of damage-associated molecular patterns (DAMPs), cytokines from the infarct area, and activated microglia combined with elevated levels of proinflammatory cytokines and/or chemokines produced from the gut immune cells. In the presence of proinflammatory microbiota (i.e., increased abundance of pathogenic microbes, such as Bacteroidetes, Proteobacteria, and clostridia), these DAMPs; pathogen-associated molecular patterns (PAMPs), such as IFN-1; and cytokines trigger innate and adaptive immune responses both in the ischemic brain region and in the periphery through the specialized pattern-recognition receptors (e.g., Toll-like receptors (TLRs)]. These processes in turn trigger the endothelial cells to express adhesion molecules, with the subsequent translocation of a large number of inflammatory and peripheral immune cells, such as macrophages or monocytes, neutrophils, dendritic cells, Th17 cells, Th1 cells, and regulatory T (Treg) cells (cells regulating the secretion of the anti-inflammatory cytokine IL-10, which suppresses post-ischemic inflammation) from the Peyer patches in the small intestine to the stroke-injured sites [120,121,122] (Figure 1).

#### 3.2.2. Adaptive Immune Signaling

During stroke-induced gut dysbiosis, reinforcement of chronic inflammation by the excessive recruitment or infiltration of peripheral immune cells (Th1 cells, Th17 cells, and monocytes from the Peyer’s patches to the infarct region), gut-derived toxic metabolites, and bacterial translocation suppresses the polarization of anti-inflammatory Treg cells [123]. Notably, dendritic cells migrate to mesenteric lymph nodes to drive differentiation of T cells into Treg cells under the influence of pro-inflammatory microbiota [44]. Reduced migration of Treg cells to the lamina propria also promotes γδ T-cell differentiation. Pro-inflammatory Th17, Th1, and γδ T-cells originating from the lamina propria of the small intestine migrate to the meninges, accelerate inflammatory damage, and increase the infarct size [78,124]. Additionally, brain-injury-derived DAMPs and the cytokine storm from activated microglia with increased amounts of IL-6, TNF-α, and IFN-γ stimulate the vagus nerve, which, in return, causes gut dysmotility, dysbiosis, increased permeability, intestinal injury, and sepsis [118].

These toxic changes in the GI tract can lead to the excessive translocation of pathogenic bacteria, bacterial toxins, and toxic metabolites, all of which in turn leak more gut inflammatory and immune cells into circulation and then to the stroke-injured sites. Furthermore, breakdown of the intestinal epithelial barrier and BBB, a decrease in enteric nerves, loss of goblet cells, and thinning of the mucus barrier bolster chronic systemic inflammation [125]. Thus, gut dysbiosis creates a vicious proinflammatory loop, which worsens the post-stroke treatment outcome.

### 3.3. Other Key Signaling Pathways in Stroke and Gut Dysbiosis 

Gut dysbiosis influences stroke pathology or prognosis mainly by affecting the key immunological signaling pathways. The three most important pathways that have serious implications in gut-dysbiosis-associated neurological diseases are:

(i) Inflammasome signaling pathway—This pathway comprises an innate immune signaling complex that responds to a variety of microbial and endogenous pathogenic signals. Inflammasome activation by SCFAs leads to IL-18 secretion, thus contributing to gut homeostasis, and was able to prevent intestinal injury in a colitis model [126]. Inflammasome-mediated gut dysbiosis impacts neuropsychiatric diseases, such as major depressive disorders. In depressed patients, hyperstimulation of the inflammasome leads to increased levels of proinflammatory cytokines [127,128].

(ii) Type I interferon (IFN-1) signaling pathway—IFN-1 is a pleiotropic and universal cytokine, playing a significant role in both innate and adaptive immunity, and thus contributes to host homeostasis. Host IFN-I can affect the composition of GM, suggesting a bidirectional link between GM and IFN-I signaling in stroke [129]. Thus, following stroke, altered IFN-1 pathways can lead to gut dysbiosis. In a murine model, Martin et al. showed that autophagy proteins suppressed the protective microbiota-dependent IFN-I signaling pathway [130].

(iii) Nuclear factor (NF)-κB signaling pathway—The NF-κB pathway is a central hub for inflammatory processes in the host. The NF-κB family comprises several key transcription factors that regulate innate and adaptive immune responses. The NF-κB pathway determines the expression of proinflammatory and proapoptotic genes and thus maintains immune homeostasis [131]. Specifically, NF-κB signaling is the main pathway affected by altered levels of gut-derived microbial products that results in gut dysbiosis and immune-derived chronic systemic inflammation [132]. Optimal activation of NF-κB activity in intestinal epithelial cells maintains the normal intestinal homeostasis by preventing pathogenic invasion and intestinal injury. Recent evidence suggests that hyperactivation of NF-κB due to gut dysbiosis results in chronic intestinal inflammatory conditions, which is reflected by excessive production of proinflammatory cytokines from lamina propria mononuclear cells (macrophages and dendritic cells) and intestinal epithelial cells [133]. In mice with antibiotic-induced gut dysbiosis, inhibition of BDNF expression and hyper-activation NF-κB signaling in hippocampal neurons leads to neuroinflammation and anxiety-like behaviour [134]. In a colitis mouse model, gut dysbiosis resulted in increased NF-κB activity in both the intestine and hippocampal regions with increased TNF-α expression, which manifests as severe memory impairment [135].

In both humans and rodent models of ischemic stroke, NF-κB signaling was shown to play a major role in tissue viability and recovery from ischemic insult [136]. In a rat model of hypertension, increased activity of matrix metalloproteinase indicated the hyper activation of NF-κB [137], thus suggesting that inhibition of the NF-κB cascade by anti-inflammatory drugs may be a promising target for the treatment of ischemia-reperfusion injury. Preclinical studies have shown that potential inhibitors of the NF-κB pathway lead to desirable outcomes in rodent models of ischemic stroke, such as reduced infarct size and inflammatory biomarkers [138,139,140].

## 4. Preclinical Studies on the Correlation of Gut Dysbiosis with Stroke

Several experimental studies in stroke-related animal models have demonstrated a direct association between dysregulated GBA signaling and gut motility, dysbiosis, intestinal permeability, inflammation, and altered immune response [44,100,141]. Multiple reports also showed that disturbance in the gut microbiota (i.e., dysbiosis) promoted the abundance of intestinal T cells that aggravates ischemic lesions [44,78]. Bacteria originating from the small intestine have been found to be a major cause of post-stroke infections in both stroke patients and mouse stroke models [142]. Unlike primary autoimmune diseases of CNS, transient brain ischemia triggers a rapid, local neuroinflammatory reaction and peripheral immune activation [117,143,144].

Reports of microbial pathogenic changes occurring in stroke animal models include a decrease in species diversity (e.g., specific alterations in *Peptococcaceae* and *Prevotellaceae*), along with increased abundance of proinflammatory microbiota (e.g., Bacteroidetes phyla, Proteobacteria phyla, and *Clostridium* species) and reduced abundance of anti-inflammatory microbiota (Firmicutes and Actinobacteria phyla). Gut dysbiosis is usually associated with changes in T-cell homeostasis and induction of a proinflammatory response by migrating T cells, T helper cells, and monocytes from Peyer patches to the peri-infarct brain region, which negatively affects post-stroke outcomes. Moreover, the presence of gut dysbiosis predisposed to the induction of larger infarct lesions and impaired recovery in an ischemic mouse model [78]. Indeed, virtually all the brain structures were found to be vulnerable to gut-dysbiosis-mediated ischemic brain injury (Figure 2).

Using human flora-associated animals (mice developed using FMT replicating a human microbial ecosystem) and germ-free (GF) mice devoid of complex microbiota, developed by growing under GF conditions or treatment with antibiotics, are considered powerful models for studying the ecosystem and influence of the human intestinal flora [146,147,148]. Both animal and human clinical studies have confirmed that FMT from healthy donors improves the cognitive and neurobehavioral functions of the recipients [149,150,151].

Induction of stroke in GF mice is associated with larger infarct areas compared with recolonized or specific pathogen-free mice, indicating that the composition of the gut bacteria plays an important role in modulating post-stroke outcome [43]. For instance, aberrant gut microbiota transferred by FMT from human hypertensive donors causes hypertension in GF mice [5]. In addition, induction of a large hemispheric lesion by proximal middle cerebral artery occlusion (MCAO) in mice (as an ischemic stroke model) caused gut dysbiosis, intestinal paralysis, increased gut permeability, a loss of cholinergic innervation in the ileum, reduction in the number of goblet cells and mucin production [109], and increased sympathetic activity [78,100,142]. Both young and aged mice [124] subjected to stroke procedures showed increased intestinal permeability, with elevated translocation of gut bacteria and bacterial toxins. Additionally, antibiotic-induced gut dysbiosis significantly reduced the survival rate of mice following the induction of stroke [152]. A study examining the role of aging-related dysbiosis in post-stroke outcomes [59] showed that aged mice (≥20 months) had a ~9-fold higher Firmicutes/Bacteroidetes ratio compared to young mice (≥3 months). Furthermore, FMT from young mice into aged mice reduced mortality, improved locomotor function and anxiety, and enhanced motor strength during the course of recovery from proximal MCAO [59]. Gut dysbiosis in genetically diabetic (*db/db*) mice exhibited increased intestinal permeability and higher plasma LPS levels, along with elevated abundance of Bacteroidetes phyla and *Enterobacteriaceae* family members (especially, *Escherichia coli*, *Klebsiella*, and *Salmonella*). There were also lower levels of Firmicutes, Actinobacteria, and Tenericutes in the diabetic mice compared to the non-diabetic mice [77,153]. Induction of focal cerebral ischemia in diabetic mice caused larger infarct volume and further enhanced the expression levels of LPS, TLR4, and inflammatory cytokines in the ischemic brain, in addition to inducing severe neurological deficits and higher mortality rates compared to those of non-diabetic mice [77].

Benakis and colleagues showed that mice with an anti-inflammatory gut microbiome (induced by antibiotics to decrease species abundance and diversity) showed reduced cerebral infarct volume (60%) and better-preserved sensory–motor function for at least 1 week following proximal MCAO [44]. Furthermore, FMT from healthy donors with minor inflammation into recipient naïve mice decreased the infarct volume by 54% under MCAO-stroke conditions [154].

Singh and his team (2016) demonstrated that FMT from the stroke donor mice (that had been previously subjected to proximal transient MCAO) to recipient mice (3 days before distal MCAO) induced a significant increase in infarct volumes and functional impairment when compared to mice receiving FMT from sham-operated mice [78]. Moreover, FMT from control mice to recipient mice (started on the day of proximal MCAO) significantly reduced gut dysbiosis and the infarct volume. Similarly, the brain of GF mice subjected to distal MCAO showed doubled lesion volumes, impaired behavioral performance, and increased levels of inflammatory markers of Th1 and Th17 T cells (mRNA of IFN-γ and IL-17, respectively) after receiving dysbiotic post-stroke cecal contents when compared with mice receiving FMT from sham mice.

In a recent study [145], GF male C57BL/6J mice receiving FMT from male mice with cerebral ischemic/reperfusion injury induced by occlusion of bilateral carotid common carotid arteries (BCCAO group) or age-matched healthy sham-operated mice (control group) showed that the gut microbial composition altered significantly in the BCCAO group when compared to the control group on day 29 after FMT. In the BCCAO and control groups, the composition of Bacteroidetes, Firmicutes, Proteobacteria, Verrucomicrobia, Actinobacteria, Tenericutes, Deferribacteres, and other phyla were 53.85% vs. 47.52%, 42.05% vs. 42.29%, 2.24% vs. 2.49%, 0.03% vs. 6.21%, 0.35% vs. 0.50%, 0.43% vs. 0.08%, 0.25% vs. 0.10%, and 0.81% vs. 0.08%, respectively, on day 15 after FMT. However, the compositions were 70.07% vs. 48.43%, 23.19% vs. 47.49%, 2.70% vs. 2.56%, 1.11% vs. 0.01%, 0.22% vs. 0.49%, 1.38% vs. 0.67%, 0.23% vs. 0.03%, and 1.09% vs. 0.33%, respectively, on day 29 after FMT, suggesting increased abundance of potentially pathogenic Bacteroidetes, Proteobacteria, and Verrucomicrobia with decreased populations of Firmicutes and Actinobacteria after FMT. Moreover, results of microbial analyses indicated that relative abundance of microbial colonization of 20 genera significantly differed between the BCCAO and control groups. These genera belonged to the phyla Bacteroidetes (7/20), Firmicutes (9/20), Cyanobacteria (1/20), and Proteobacteria (3/20). These results showed that microbial colonization from the BCCAO group negatively impacted the gut microbiota of GF mice. Resting-state functional magnetic resonance imaging (MRI) performed to assess functional connectivity (FC) between specific regions of the brain showed a significant decrease in the FC in the cingulate cortex and thalamus of BCCAO rats compared to the control rats (Figure 3). 

Acute ischemic stroke in mice was shown to upregulate the protein expression of autophagy markers, such as light-chain 3-II Beclin-1 and autophagy-related gene (*Atg*) 12, with increased levels of ROS, NADPH oxidase 2/4 (NOX2/4), lipid peroxide malondialdehyde, homocysteine, and free fatty acids (triglyceride and cholesterol), reflecting ischemia-induced neuronal damage [155,156,157,158]. However, the levels of total antioxidant capacity and activity of superoxide dismutase (SOD) and reduced glutathione (GSH) were found to be decreased in brain tissues of the stroke mice [158]. Altogether, these preclinical studies reveal that the gut microbiota can be manipulated in a manner to either improve or worsen post-stroke outcomes. In general, a healthy gut microbiota (eubiosis) stabilizes the gut wall and regulates low-degree inflammation, providing a protective defence against intestinal-barrier dysfunction and infections. However, conditions that produce inflammation and/or dysbiosis in the gut are found to accelerate brain injury and negatively affect the prognosis in stroke models through the GBA.

## 5. Clinical Studies on Stroke-Associated Gut Dysbiosis

Recent studies have revealed that changes in gut microbiota ecology are related to many diseases states, including cerebrovascular diseases [159,160]. Although a few clinical investigations have been reported on stroke-induced gut dysbiosis, they unambiguously showed the occurrence or presence of gastrointestinal symptoms following the onset of stroke [2,110]. Furthermore, clinical studies conducted to delineate the crucial link between gut microbiome and stroke have confirmed the significant changes in the diversity and abundance in faecal microbial samples of stroke patients [111].

A clinical study [110] showed that patients with large-artery atherosclerotic stroke or transient ischemic stroke attacks have distinct gut microbial composition, along with increased abundance of opportunistic pathogens (e.g., *Megasphaera, Enterobacter, Desulfovibrio,* and *Oscillibacter)* and decreased abundance of commensal or beneficial genera (e.g., *Prevotella*, *Bacteroides*, and *Faecalibacterium*) compared to asymptomatic controls with or without carotid atherosclerotic plaques. In addition, blood TMAO levels were reported to be lower in patients with large-artery atherosclerotic stroke or acute brain ischemia.

A Japanese cohort study [2] revealed that ischemic stroke was associated with increased abundance of *Lactobacillus ruminis Atopobium cluster* and reduced abundance of *Lactobacillus sakei* subgroup, hypertension, independent of age, and T2DM when compared to control subjects. Ischemic stroke is associated with decreased acetic acid levels (negatively correlated with levels of low-density lipoprotein cholesterol and glycated haemoglobin) and with increased valeric acid levels (positively correlated with the level of white blood cell counts and high-sensitivity C-reactive protein), indicating that ischemic stroke induces altered host metabolism and systemic inflammation. Patients with coronary artery disease showed a dysbiosis pattern with enrichment in *Enterococcus* and *Escherichia-Shigella*, as well as a reduced abundance of *Faecalibacterium*, *Roseburia*, *Subdoligranulum*, and *Eubacterium rectale* [161]. Metagenomic analyses of the faecal samples from patients with chronic heart failure revealed lower abundance of *Faecalibacterium prausnitzii* and increased abundance of *Ruminococcus gnavus* [162].

A clinical study in China [113] showed that the microbial α-diversity and composition were similar between cerebral ischemic (CI) stroke patients and healthy controls, but gut microbiota of stroke patients had an increased number of SCFA-producing taxonomies, such as *Akkermansia, Odoribacter, Ruminococcaceae_UCG*_005, and *Victivallis*. Furthermore, the abundance of the genera norank_f*_Ruminococcaceae, Christensenellaceae_*R-7_*group,* and *Enterobacter* positively correlated with stroke severity, whereas the abundance of *Christensenellaceae*_R-7group positively correlated with the clinical outcome of CI patients. Within the CI group, *Pyramidobacter*, *Enterobacter*, and *Lachnospiraceae*_UCG_001 were increased in patients with mild CI stroke, whereas the genera *Ruminococcaceae*_UCG-005, *Ruminococcaceae*_UCG002, *Christensenellaceae*_R-7_group, and norank_f_*Ruminococcaceae* were increased in severe stroke patients, suggesting that alterations in intestinal microbiota correlate directly with severity of CI in humans (Figure 4 and Figure 5).

Another clinical study [163] investigated the gut microbiome of 141 participants (aged 60 years or above) by classifying them into low-, medium-, and high-risk groups based on the presence of known risk factors, including diabetes mellitus, hypertension, dyslipidemia, atrial fibrillation, smoking, overweight, sedentary lifestyle, and family history of stroke. The key results showed that individuals at a high risk of stroke showed elevated levels of gut opportunistic pathogens (e.g., Betaproteobacteria*,* Desulfovibrionaceae*, Enterobacteriaceae*, Actinomycetaceae, Veillonellaceae, *Veillonella*, *Megasphaera*, *Acidaminococcus*, and *Sutterella*) and lactate-producing bacteria (e.g., *Lactobacillus* and *Bifidobacterium*) with lower abundance of butyrate-producing bacteria (e.g., *Lachnospiraceae* (*Lachnospira*, *Roseburia*, and *Oscillospira*) and Ruminococcaceae (*Faecalibacterium*)) compared to those of the individuals with a low-risk of stroke. Additionally, the relative abundance of Proteobacteria, Lactobacillales, Bacilli, Streptococcaceae, and Fusobacterium was also higher in the high-risk group than that in the low-risk group. Interestingly, an increase in the abundance of *Lactobacillus* and *Bifidobacterium* was reported in high-risk patients. Similarly, faecal butyrate levels were also lower in the high-risk group than the low-risk group.

A clinical study demonstrated the presence of gut microbial dysbiosis in individuals with acute ischemic stroke when compared to healthy controls by employing a Stroke Dysbiosis Index model (SDI) for measuring the gut microbial dysbiosis in stroke subjects [111]. The results indicated that 18 genera of gut microbiome differed significantly between stroke patients and healthy individuals. The SDI correlated positively with brain injury and poor functional outcomes. Patients with higher SDI had enrichment of *Enterobacteriaceae*, *Oscillospira*, *Parabacteroides*, and Bacteroidaceae and reduced abundance of butyrate-producing *Faecalibacterium*, Clostridiaceae, and *Lachnospira*. Intriguingly, FMT from patients with higher SDI to naïve mice subjected to stroke procedures showed larger infarct volumes and severe neurological deficits, along with a higher number of proinflammatory (IL17+) γδ T cells and a lower number of (CD4+CD25+) T cells in the spleen than those of the corresponding mice after FMT from patients with a low SDI.

A recent clinical study [112] revealed a lack of SCFA-producing bacteria, including *Roseburia*, *Blautia*, *Bacteroides*, *Faecalibacterium*, *Lachnospiraceae*, and *Anaerostipes*, and an increased abundance of opportunistic pathogens, such as *Akkermansia*, Lactobacillaceae, *Enterobacteriaceae*, and Porphyromonadaceae, confirming gut dysbiosis in patients with acute ischemic stroke, especially in those with increased severity, when compared to the healthy controls. Patients with higher stroke severity had reduced levels of faecal SCFAs. An increased ratio of Bacteroidetes/Firmicutes was observed in acute ischemic stroke patients compared to healthy controls [112].

Karlsson and colleagues (2012) demonstrated that patients with symptomatic atherosclerotic plaques showed abundant genera of *Bacteroides*, *Ruminococcus*, *Eubacterium*, and *Faecalibacterium* and low abundance of butyrate-producing bacterium SSC/2. The shotgun sequencing of the gut metagenome showed that the genus *Collinsella* was enriched in stroke patients, whereas *Eubacterium* and *Roseburia* were enriched in healthy controls. Moreover, gut metagenomes showed that genes associated with peptidoglycan biosynthesis were elevated, whereas genes encoding phytoene dehydrogenase K10027 were decreased in patients, along with reduced levels of antioxidants, such as β-carotene, when compared to control subjects. These results indicate that gut microbiota may contribute to symptomatic atherosclerosis by priming the associated inflammation and innate immune system pathways [114].

Taken together, these clinical studies suggest that toxic changes in gut microbial composition and their abundance correlate directly with the magnitude of stroke severity. However, the timing of microbial changes and their impact on the onset of ischemic stroke in patients remain poorly understood.

## 6. Novel Therapeutic Strategies in the Modulation of Intestinal Microbiota for the Prevention and Treatment of Stroke

The complexity of neural, immune, hormonal, and (neuro) endocrine interactions between the GI tract and gut microbiota, cardiovascular system, and the brain presents therapeutic challenges. The results of animal and human clinical studies reveal that gut dysbiosis can be a potential risk factor for onset, severity, and outcome in stroke patients. Thus, new therapeutic strategies seek to restore a healthy gut microbiome for the prevention and treatment of stroke [164,165]. Modulation of gut microbiota towards a healthy state by nutritional interventions using prebiotics, probiotics, synbiotics, and antibiotics, and by FMT from normal donors is likely to help prevent the pathogenesis of stroke [166,167,168] and cognitive and behavioral impairments associated with aging and stroke [169,170].

Clinical studies showed that treatment with broad-spectrum antibiotics reduced the risk of infection in stroke patients [118,171]. However, antibiotic treatment studies on mouse stroke models have produced inconsistent and discrepant results. Oral administration of a non-absorbable antibiotic (polymyxin B at dose of 1 mg dissolved in 0.2 mL of distilled water once daily given via oral gavage for one week) improved gut microbiome and improved stroke outcomes. These beneficial effects are concordant with a reduction in LPS levels and neuroinflammation in the ischemic brain [77]. In contrast, other experimental studies demonstrated that usage of broad-spectrum antibiotics caused significant damage to gut microbiota and worsened stroke outcomes. The results of the Preventative Antibiotics in Stroke Study (PASS) in adult patients with ischemic or haemorrhagic stroke did not support the preventive use of antibiotics (e.g., 2 g of ceftriaxone given intravenously every 24 h for 4 days). Although the rate of infection was reduced, there was no improvement in functional outcome, duration of hospitalization, or reduction in mortality [172]. Thus, the beneficial effects of using antibiotics against post-stroke outcomes need further studies.

### 6.1. Probiotics/Prebiotics/Synbiotics

Probiotics and prebiotics have been shown to help establish eubiosis in stroke patients. However, the precise mechanisms for their beneficial effects remain largely unknown and are the topic of intense study. Probiotics refer to the living microorganisms that, when ingested by humans or animals, can confer a health benefit [90,173]. Prebiotics refer to non-digestible fibres or antioxidant compounds that are selectively metabolized in the small intestine and promote the growth of symbiotic species [90]. Synbiotics are defined as synergistic mixtures of probiotics and prebiotics that confer benefit to the host by selectively promoting the growth of beneficial microbes [174]. Supplementation with probiotics or prebiotics may improve post-stroke outcomes by reducing gut leakiness and plasma LPS levels [175,176], as these supplements improved lipid and glucose metabolism in overweight people and those with diabetes mellitus [177].

Another study showed that probiotic supplementation reduces blood pressure, one of the major risk factors for stroke [178]. Supplementation with probiotics alters the composition and abundance of the gut microbiome by modulating cytokine release and neuroinflammatory response. Therefore, probiotics represent another additional therapeutic approach for the management of cardiometabolic disorders, including arterial hypertension and acute stroke [179,180,181]. The known mechanisms of action of probiotics include suppression of TNF-α and free radicals via the TLRs in the gut epithelium [182], reduction in TMAO levels, an increase in the production of brain-derived neurotrophic factor (BDNF), inhibition of apoptosis, and increased abundance of some symbionts [183,184]. Pretreatment with *Clostridium butyricum* attenuated hippocampal apoptosis and improved neurological deficit scores in mice following BCCAO. Intragastric pretreatment with 1 × 10^9^ CFU of *C. butyricum* (200 µL once daily for 2 successive weeks) before bilateral common carotid artery occlusion (for 20 min) showed increased butyrate levels and reduced oxidative stress in the brain of male ICR mice, indicating the neuroprotective effects [103]. In a rat model of transient focal cerebral ischemia (proximal MCAO), sodium butyrate (intracerebroventricular injection of 2 µL/min by a micro infusion pump) improved BBB integrity and decreased the activity of matrix metalloproteinase [185]. In a mouse model of stroke, oral pretreatment with a probiotic bacterial mixture *(Bifidobacterium breve, Lactobacillus casei, Lactobacillus acidophilus,* and *Lactobacillus bulgaricus of 10^7^ CFU/mL* via oral gavage daily for 14 days) reduced cerebral ischemia by 52% and significantly improved the neurological outcome [182].

Prebiotics help restore the gut microbiota and negatively correlate with risk of cardiometabolic diseases. Diets rich in fibre and plant polyphenols alter gut microbial ecology to eubiosis [45]. Fibres and polyphenols are converted into biologically active compounds by gut microbes and maintain colon-metabolic homeostasis while they serve as antioxidants to neutralize ROS and reactive nitrogen species (RNS). For instance, plant-derived polyphenols, which are rich in antioxidants, provide ameliorative effects on vascular endothelial cells via reducing oxidation of low-density lipoproteins [186,187]. Emerging evidence shows that high intake of dietary fibre reduces blood pressure in patients with hypertension [188]. A high-fibre diet increases the number of acetic-acid-producing bacteria, which leads to a reduction in blood pressure in hypertensive mice [189]. The latter diet also enhances the abundance of butyrate-producing bacteria, which were associated with reduced gut leakiness, endotoxemia, systemic inflammation, and susceptibility to atherosclerotic lesions in a murine model [190]. A mouse model of stroke (induced by photothrombotic stroke surgery) showed that lower plasma SCFA levels are correlated with worse outcomes, whereas supplementation with SCFAs reduced motor deficits by modulating systemic and brain resident immune cells [191]. Overall, these results clearly show that greater dietary fibre intake inversely correlates with the development and treatment outcome of cardiovascular diseases and stroke [192]. Nonetheless, it should be acknowledged that there is significant methodological heterogeneity across studies describing the impact of probiotics, prebiotics, and synbiotics on stroke. The use of different species, methods of administration, and dosages can lead to different clinical effects, and thus, future studies will be required to standardise methodologies in well-designed large randomised controlled trials.

### 6.2. Fecal Microbiota Transplantation (FMT)

FMT therapy refers to an administration of stool from healthy donors via enema, nasoenteric, nasogastric, or endoscopic (upper endoscopy, sigmoidoscopy colonoscopy) routes or oral capsules [193,194] that allow establishment of a new gut microbiota community. Studies have reported that the effects of FMT can be variable, depending on the study design, the usage of different donors (i.e., quality control), delivery routes, and the use of different antibiotics. FMT can, however, be used as an adjuvant therapy or in combination with other treatment methods to modulate the gut microbiome toward a healthy eubiosis state. Combined treatment methods are found to promote healthy gut microbiota and also repair gut damage, including intestinal permeability. Moreover, FMT methods are cost-effective [195,196,197]. FMT has evolved as a new potential strategy for restoration of gut microbial dysbiosis, involved in complex pathologies of several clinical conditions, including metabolic syndrome, alcoholic liver disease, autoimmune disorders, and cardiovascular and neurological diseases [195,196,198]. Recent studies in animal models also showed the benefits of FMT in improving post-stroke outcomes [2,199].

In an MCAO-induced stroke model [78], FMT not only reduced lesion size but also increased Foxp3+ Treg cell counts in the ischemic cerebral hemispheres and the peripheral immune system. Metagenomics analysis showed that FMT normalized stroke-induced gut dysbiosis and partially restored microbial diversity in mice with severe infarcts. Additionally, more in-depth analysis demonstrated an increase in taxonomic abundance of eubacterial phyla, which were reduced in stroke groups compared to sham-operated groups. However, a model of lymphocyte-deficient *Rag1−/−* mice showed that FMT had no effect on lesion size, indicating that gut-microbiota-mediated neuroprotective effects on brain injury are mediated by modulating lymphocytes or immune cells [78], similar to other recent results [191]. In a rat model of MCAO, intragastrical administration of non-absorbable antibiotics (vancomycin (100 mg/kg), neomycin sulphate (200 mg/kg), metronidazole (200 mg/kg), and ampicillin (200 mg/kg) daily for 4 days) improved neurological functions and reduced the cerebral infarct volume. An FMT (rich in SCFA-producing bacteria) intervention, along with butyric acid supplementation (30 mg/kg) given intragastrically once a day for 14 days) were effective in stroke treatment by restoring the normal gut microbiota and enhancing α-diversity, along with significant enrichment of beneficial bacteria, such as *Lactobacillus, Butyricicoccus,* and *Meganonas,* with concomitant reduction of opportunistic pathogens, such as *Alistipes, Klebsiella, Bacteroides, Shuttleworthia, Fusobacterium, Haemophilus, Faecalibacterium, Proteus*, and *Papillibacter* [200].

Antibiotics (neomycin (450 mg/L) and Polymyxin B (150 mg/L) mixed with drinking water) also significantly relieved cerebral edema, reduced serum total cholesterol levels, and restored the integrity of the intestinal wall with the elevated faecal levels of acetic, is butyric, butyric, valeric and isovaleric acids. FMT also significantly reduced the infarct volume, neurological deficit, serum total cholesterol, and triglyceride levels; eliminated brain edema; restored intestinal permeability; and enhanced isobutyric, butyric, and isovaleric acid levels in faeces. Moreover, butyric acid supplementation significantly reduced neurological impairment, cerebral infarct volume, serum total cholesterol, triglycerides and fibrinogen levels, whole blood viscosity, and intestinal permeability, whereas it relieved cerebral edema [200]. These studies suggest that FMT, with its high content of SCFA-producing bacteria (particularly butyric acid), promotes a positive clinical outcome in cerebral ischemic stroke by reducing or preventing gut dysbiosis, intestinal barrier dysfunction, neurological impairment, cerebral infarction volume, blood lipid levels, cerebral edema, neurotoxicity, neuroinflammation, and the risk of thrombosis.

Microbial transplantation studies have shown a positive correlation between gut-microbe-dependent TMA/TMAO production atherosclerosis plaque formation and elevated susceptibility for vascular thrombosis [70]. Although the aforementioned studies suggest potential benefits of FMT-mediated reprogramming of the gut microbiota for cerebrovascular disease, the rationale for the clinical application of FMT is mainly based on animal models. Large-scale randomized double-blind controlled clinical trials are still needed to clarify the role of FMT in preventing or treating patients with stroke and other neurological diseases [201].

### 6.3. Natural Bioactive Compounds Used in Stroke Treatment

Several preclinical and clinical studies have clearly stated that natural bioactive compounds, such as legumes, fruits, vegetables, wine, olive oil, and whole grains, consumed as food in diet or as dietary supplements, or other plant-derived compounds, possess cardioprotective effects. Many in vitro studies have suggested that several molecules with distinct chemical structures, such as polyphenolic compounds, peptides, oligosaccharides, vitamins, and n-3 fatty acids, are found to be potent cardioprotective agents [202,203]. Among them, the most efficient bioactive compounds exhibiting significant cardioprotective effects are long-chain omega-3 polyunsaturated fatty acids, including plant-derived α-linolenic acid, and fish-oil-derived eicosapentaenoic acid and docosahexaenoic acid [204,205,206,207].

The principle bioactive molecules found in teas are polyphenols, such as catechins in green tea and the aflavins in black tea. Both in vivo and in vitro models of cerebral ischemic/reperfusion (I/R) have shown the beneficial effects of aflavin by obliterating miRNA-128-3p-induced nuclear factor erythroid 2-related factor 2 (Nrf2) suppression and thus decreasing oxidative stress [208]. Another catechin found in green tea, (-)-epigallocatechin-3-gallate, treatment reduced neurological deficit, decreased infarct volume, promoted angiogenesis, upregulated vascular endothelial growth factor receptor 2 signaling pathway, and increased Nrf2 nuclear levels [209]. Pretreatment with (-)-epicatechin before permanent MCAO reduced infarct volume and ameliorated neurological deficits in Nrf2/2 knockout (*Nrf2−/−)* mice compared with wild-type controls [210]. Nomilin from citrus fruits protected SH-SY5Y cells against oxygen glucose deprivation (OGD) and in a rat/mouse model of cerebral I/R by reducing the infarct volume, neurologic score, and oxidative stress [211,212]. Both in vitro and in vivo models have shown that naringenin and nobiletin in citrus fruits reduced neurological deficits, brain edema, and infarct volume and exerted strong antioxidant action through the involvement of the Nrf2 signaling pathway [213,214,215]. Spices have been used as medicines for several centuries. Mainly, garlic, turmeric, chili peppers, and rosemary contain *S*-allyl cysteine, diallyl trisulfide, dihydrocapsaicin, curcumin and rosmarinic acid, respectively [216]. In vivo models in both mice and rats subjected to I/R injury have proven the neuroprotective effects of these spices [217,218,219,220,221]. The presence of oleuropein and hydroxytyrosol in olive oil account for the most abundant polyphenols, with excellent free-radical scavenging properties in experimental models [222]. Anthocyanins, a large subgroup of flavonoids, prevent oxidative injury in the H9C2 rat cardiomyocyte cell line [223].

Among the fruit-based bioactive compounds, *Lycium barbarum* fruits (known as Goji berries) contain monoterpenes, flavanols, phenolic acids, and lyceum amides [224]. Lycium amide A provided protective effects against stroke [225]. In vivo studies investigating plant-based bioactive compounds showed that treatment with a dichloromethanic fraction of mango leaves (100 mg/kg twice daily for 30 days) (Ronchi et al.) reduced cardiac hypertrophy and increased the ratio of heart weight to body weight in spontaneously hypertensive rats [226]. Mangiferin in mango and papaya improved neurological score and decreased infarct volume and edema in rats subjected to cerebral I/R injury [227]. Interestingly, both in vitro astrocytic cultures and experimental stroke models, preconditioning with resveratrol, present in grapes, revealed neuroprotective effects [228]. Procyanidin B are the predominant polyphenols found in berries, cereals, legumes, nuts, chocolates, and wines [229]. Specifically, procyanidin B2 in cocoa, grapes, and apples was found to reduce infarct size, brain edema, and neurological deficits after MCAO by preventing BBB damage, attenuating tight junction degradation, and counteracting oxidative stress through the Nrf2 pathway [230]. The antioxidant properties of these natural bioactive polyphenols provides neuroprotection directly or indirectly by potentially improving the gut dysbiosis associated with stroke models. The main compounds, such as polyphenols and flavonoids, restore the commensal microbes and thereby have been shown to improve post-stroke health.

## 7. Conclusions and Perspectives

In the present review, we have briefly discussed the association of gut dysbiosis with the pathogenesis and treatment outcome of stroke. Clinical and preclinical studies over the last decade have demonstrated a compelling link between gut dysbiosis and the development of well-known stroke risk factors, such as dyslipidemia; insulin resistance; obesity; hypertension; T2DM; and cardiovascular, cerebrovascular, and neurological disorders. Recent research findings have confirmed a positive correlation between gut dysbiosis and poor post-stroke outcomes. The pathophysiological mechanisms underlying stroke-induced gut dysbiosis include breakdown of the intestinal epithelial barrier, tight and adherent junction proteins, gut dysmotility, altered mucus secretion, loss of goblet cells, alterations of local immune homeostasis, elevated LPS levels, and intestinal inflammation. Consequently, immune-driven systemic inflammation or endotoxemia takes place, and this event leads to elevated inflammation, breakdown of BBB, and neurotoxicity, marked by increased production of ROS, RNS, TMAO, and NOX2/4 and neuroinflammation. The damaging consequences are caused by high levels of LPS, C-reactive protein, TLR4, bacterial toxins, toxic metabolites, abnormally activated immune cells, and pro-inflammatory cytokines in the ischemic brain region. These changes result in larger infarct size, neuronal death, synapse loss, and glial dysfunction, which accounts for poor outcomes. Inflammatory signals from the infarct site further enhance gut dysbiosis, thus resulting in a proinflammatory loop. To break the vicious proinflammatory cycle, the gut microbiome has been considered a promising target in the treatment and prevention of stroke in experimental and clinical studies. Deleterious effects of gut dysbiosis can be prevented by pharmacological and non-pharmacological methods, such as dietary interventions, antibiotics, prebiotics, probiotics, and synbiotics, as well as FMT. Several studies have confirmed that gut microbiome dynamics can be regulated toward a healthy state by restoring the beneficial microbial population and elevated levels of SCFAs (especially butyric acid). In cerebral ischemic stroke, supplementations rich in SCFA-producing bacteria have been shown to significantly reduce intestinal leakage and inflammation; decrease the abundance of pathogenic bacteria (e.g., *Bacteroides*, *Klebsiella*, and *Haemophilus*); and increase the population of beneficial bacteria (e.g., *Lactobacillus*, *Butyricicoccus*, and *Meganonas*), which in turn inhibit the apoptosis of neural cells, oxidative stress, and cerebral infarction volume, leading to prevention of neurobehavioral impairments. Conclusively, FMT with enriched SCFA-producing bacteria and butyric acid supplementation is found to be an effective treatment for cerebral ischemic stroke, although additional large-scale randomized clinical studies are needed in the future to demonstrate the efficacy and safety of FMT in treating patients with stroke or other cardiovascular complications. It will be important to identify conserved and reliable biomarkers of the gut microbiome to help with the design of next-generation microbiome-based therapeutics for the early prevention of neurodegenerative diseases, stroke, and post-stroke neurobehavioral deficits or dementia, as well as the monitoring of disease progression and the efficacy of therapeutic agents. These efforts will be greatly facilitated by longitudinal human clinical trial work, which shuld assess the effects of these microbial drugs on stroke through the gut-microbiota–brain axis.

## Figures and Tables

**Figure 1 cells-11-01239-f001:**
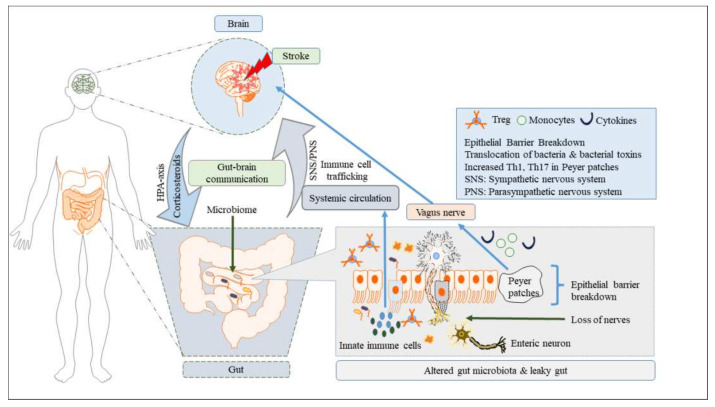
Effects of stroke on the gut–brain axis. Following cerebral stroke, gut dysbiosis causes loss of enteric nerves, increased intestinal-barrier permeability, reduced mucus production, loss of goblet cells, thinning of the mucus barrier, and increased sympathetic activity in the intestinal wall, all of which contributes to intestinal inflammation and an exaggerated immune response. These events in turn disrupt intestinal and systemic immune homeostasis, resulting in poor stroke treatment prognosis.

**Figure 2 cells-11-01239-f002:**
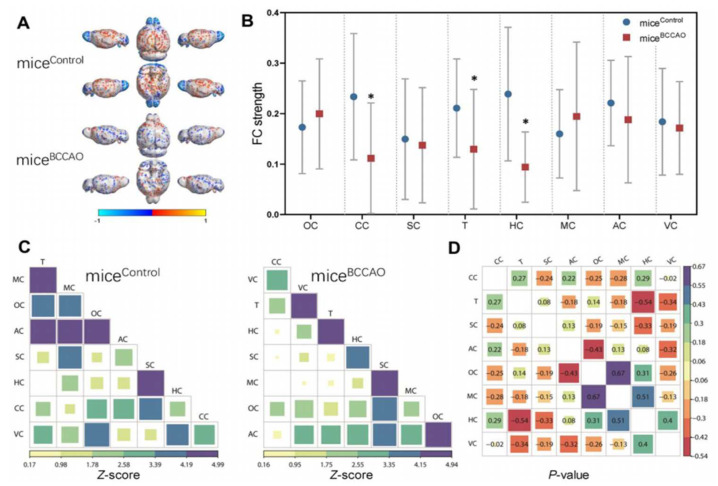
Effect of gut dysbiosis following cerebral ischemic/reperfusion injury on brain structure and function. The eight brain-specific regions, such as the orbitofrontal cortex (OC), somatosensory cortex (SC), cingulate cortex (CC), hippocampus (H), motor cortex (MC), thalamus (T), auditory cortex (AC), and visual cortex (VC), in mice subjected to cerebral ischemic/reperfusion injury induced by occlusion of bilateral carotid common carotid arteries (BCCAO group) or sham-operation (control group). (**A**) The functional connectivity between specific regions of interest of both control and BCCAO groups is shown in the virtual graphics. (**B**) Mean functional connectivity strength in brain network was measured using two-way repeated-measures ANOVA with Tukey multiple comparisons as post hoc test. (**C**) The mean functional connectivity matrices show the strength of functional connectivity between pairs of brain regions in the normal and stroke groups. (**D**) Correlation analyses in the regions (8) of interest in animal brain; the colour scale indicates the functional connectivity strength. * *p* < 0.05. The figure is reused as per journal copyright permission [145].

**Figure 3 cells-11-01239-f003:**
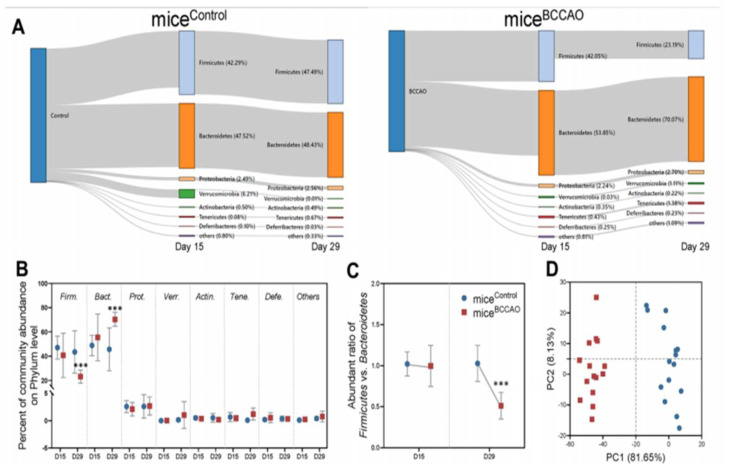
(**A**,**B**) Metagenomic analysis of mice in BCCAO and control groups showed significant changes in gut microbial composition. (**C**) Firmicutes/Bacteroidetes ratio was reduced in BCCAO vs. control groups on day 29 after FMT. (**D**) The effects of baicalin treatment on the gut microbial populations at the phylum level showed that the plot of principal component 1 against principal component 2 formed a distinct cluster. Phylum clustering in BCCAO group was different from that in the control group. *** *p* < 0.01. The figure is reused as per journal copyright permission [145].

**Figure 4 cells-11-01239-f004:**
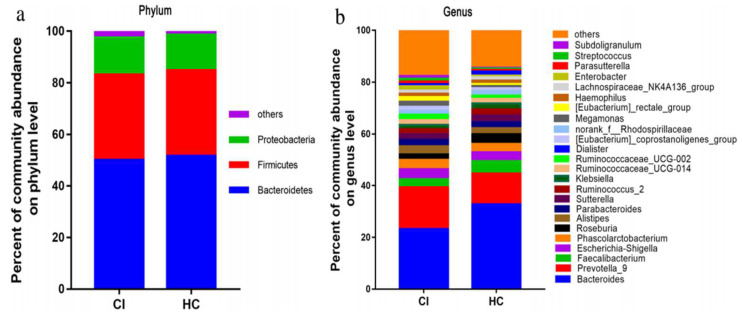
The relative taxa abundance between cerebral ischemic stroke patients (*n* = 30) and healthy controls (*n* = 30) belong to three phyla: Bacteroidetes, Firmicutes, and Proteobacteria (**a**) and to 25 genera that comprise up to 80% of the total microbiota, such as *Bacteroides*, *Prevotell*a, *Faecalibacterium*, *Escherichia/Shigella*, *Phascolarctobacterium*, and *Roseburia* (**b**). The figure is reused as per journal copyright permission [113].

**Figure 5 cells-11-01239-f005:**
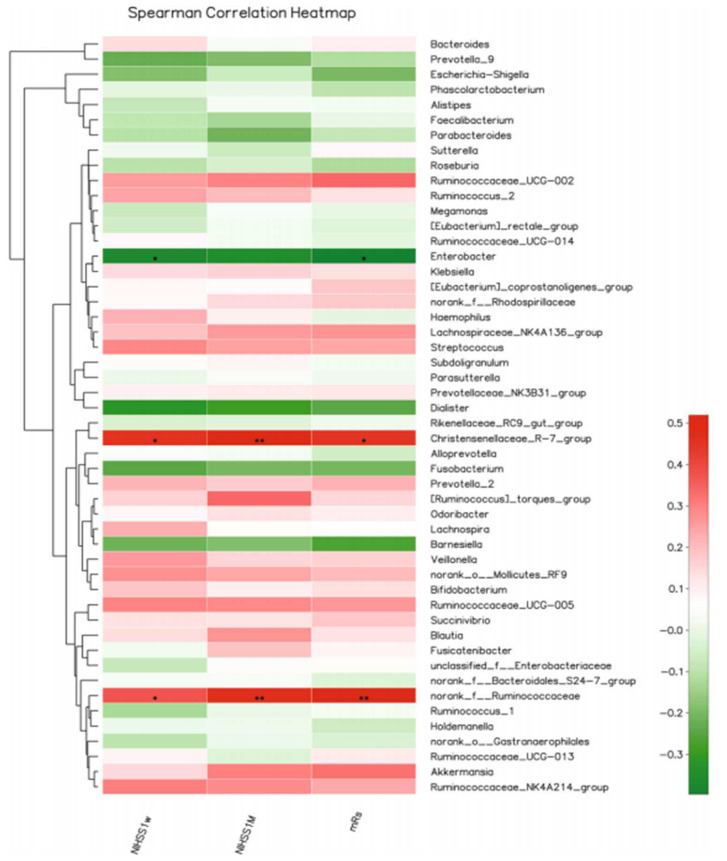
Heat map of Spearman correlation coefficient analysis, including the National Institutes of Health Stroke Scale (NIHSS) after 7 and 30 days and modified Rankin scale (mRS), was used to explore the relationship between the gut microbiota and severity or outcome of cerebral ischemic stroke patients. * *p* < 0.05, ** *p* < 0.01. The figure is reused as per journal copyright permission [113].

**Table 1 cells-11-01239-t001:** Gut-derived bioactive metabolites and associated microorganisms.

Gut-Derived Metabolites	Microorganisms	References
Acetate and propionate	Bacteroidetes (Gram-negative microorganisms), mainly *Bacteroides thetaiotaomicron* and *Bifidobacterium* species (Phylum: Actinobacteria).	[74]
Butyrate	Firmicutes (Gram-positive microorganisms), particularly *Faecalibacterium prausnitzii* (Phylum: Firmicutes), *Clostridium leptum* (Family: Ruminococcaceae), and *Eubacteriumrectale* and *Roseburia* species (Family: *Lachnospiraceae*).Other potential butyrate producers include *Eubacteriumhallii* and *Anaerostipes* spp. and members of the phyla Actinobacteria, Bacteroidetes, Fusobacteria, Proteobacteria, Spirochaetes, and Thermotogae.	[75,76]
Lipopolysaccharide	Gram-negative members of *Enterobacteriaceae*, such as *Escherichia coli*, *Klebsiella*, and *Salmonella*.	[77,78]
Neurotransmitters (acetylcholine, GABA, 5-HT, glutamate, dopamine, and noradrenaline)	*Lactobacillus* species secrete acetylcholine and GABA; *Bifidobacterium* species produce GABA; *Escherichia* produce norepinephrine, 5-HT, and dopamine; *Streptococcus* and *Enterococcus* produce 5-HT; and *Bacillus* species produce norepinephrine and dopamine.	[57,79]
Gut hormones (cholecystokinin, glucagon-like peptide-1, peptide YY, glucose-dependent insulinotropic polypeptide, or gastric inhibitory polypeptide and 5-HT (acts as a local hormone in the gut and as neurotransmitter in the brain)	Indigenous spore-forming microbes from *Clostridial* species, *Corynebacterium* spp., *Streptococcus* spp., and *Escherichia coli* synthesize 5-HT; *Odoribacter*, *Akkermansia*, *Ruminococcaceae*_UCG_005, and *Victivallis* are well-known producers of SCFAs that regulate the released gut hormones in response to nutrients by enteroendocrine cells.	[80,81,82,83]
Trimethylamine-N-oxide (TMAO)	Gut microbes *Anaerococcushydrogenalis, Clostridium asparagiforme, Clostridium hathewayi, Clostridium sporogenes*, *Edwardsiellatarda*, *Escherichia fergusonii*, *Proteus penneri*, and *Providencia rettgeri* metabolize dietary choline, L-carnitine, and betaine to form trimethylamine and TMAO.	[84]

## Data Availability

The data that support the findings of this study are available in standard research databases, such as PubMed, ScienceDirect, and Google Scholar, and/or on public domains that can be searched with either key words or DOI numbers.

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
