# Peer review of "The Influence of Gut Dysbiosis in the Pathogenesis and Management of Ischemic Stroke"

_cells, 2022, doi:10.3390/cells11071239_

Round 1
Reviewer 1 Report
The review by Chidambaram et al. reports recent evidence on the influence of gut dysbiosis in the pathogenesis of ischemic stroke.
The text is plenty of typos (lack for spaces and others) and requires a careful review. Above this, I think the paper is really useful to get a wide overview on the presence of specific taxa in this pathologic status and in general on gut-brain axis.
Figures and tables well explain the aims of the different sections.
Author Response
Comments and Responses
Title : The influence of gut dysbiosis in the pathogenesis and management of Ischemic Stroke
Reviewer -1
The review by Chidambaram et al. reports recent evidence on the influence of gut dysbiosis in the pathogenesis of ischemic stroke.
- The text is plenty of typos (lack for spaces and others) and requires a careful review. Above this, I think the paper is really useful to get a wide overview on the presence of specific taxa in this pathologic status and in general on gut-brain axis
Response – We all authors sincerely thank the reviewer for kindly reviewing the manuscript. We have now gone through the manuscript in depth and corrected all typos, spacing, and grammatical issues in the revised manuscript.
- Figures and tables well explain the aims of the different sections.
Response – We also sincerely appreciate the reviewer’s comment to improve the presentation style of our figures and tables.
Reviewer 2 Report
This review entitled “The influence of gut dysbiosis in the pathogenesis and management of Ischemic Stroke” is interesting, It described one key therapeutic target for the effective management and treatment of ischemic stroke.But I have some minor concerns regarding this manuscript.
- Pay attention to the correct use of spaces. Andspelling mistakes occurred throughout the whole manuscript.
- What doesthe mean of “(?)” in abstract?
- Line 63-66, The causality of this sentence is unreasonable. It is recommended that the author rewrite this sentence.
- Line 94-95, The author is advised to rewrite the sentence to make the context more coherent.
- Table 1, a line divider should be added to separate out the gut-derived metabolites. It is proposed to add the role of Gut-derived metabolites in the host according the references in the Table 1.
- Part 2.3.1, It is recommended to enumerate these pathways in detail of brain to gut signaling or top-down.
- Part 3.1, It is recommended that the authors provide more detailed description of the role of gut dysbiosis in stroke.
- Part 4, “Preclinical Studies on the Correlation of Gut Dysbiosis with Stroke”, A lot of clinical cases were described to prove the changes of gut microbiota in Stroke. These changes are easy to observe, Can you explain these connections using further mechanismsor pathways?
- the description" therapeutic Strategies of probiotics in the Modulation of Intestinal Microbiota for the Prevention and Treatment of Stroke", There are many kindsof probiotics, such as Bacillus, Lactobacillus, etc. Can they all as therapeutic strategies in the modulation of intestinal microbiota for the prevention and treatment of stroke? In addition, it seems that different kinds of species, methods of using, and dosages of probiotics will lead to different clinical effects, authors should describe this strategy in detail and accurately in this section for guidance.
Author Response
Comments and Response
Title : The influence of gut dysbiosis in the pathogenesis and management of Ischemic Stroke
Reviewer -2
This review entitled “The influence of gut dysbiosis in the pathogenesis and management of Ischemic Stroke” is interesting, it described one key therapeutic target for the effective management and treatment of ischemic stroke. But I have some minor concerns regarding this manuscript.
- Pay attention to the correct use of spaces. And spelling mistakes occurred throughout the whole manuscript.
Response – We thank this reviewer’s comment and have now checked through the manuscript again and corrected any obvious mis-spelling, grammatical and spacing issues.
- What does the mean of “(?)” in abstract?
Response – The symbol “(?)” was a typographic error and we now remove it in the revised manuscript.
- Line 63-66, The causality of this sentence is unreasonable. It is recommended that the author rewrite this sentence.
Response – As per reviewer comment, the sentence in the line 63-66 is addressed and re-written in the revised manuscript.
- Line 94-95, The author is advised to rewrite the sentence to make the context more coherent.
Response – As per reviewer suggestion, the sentence in the line 94-95 is changed to be coherent with the overall context in the revised manuscript.
- Table 1, a line divider should be added to separate out the gut-derived metabolites. It is proposed to add the role of Gut-derived metabolites in the host according the references in the Table 1.
Response – As suggested by the reviewer, a line divider for each item is added in Table.1 of the revised manuscript. The role of gut-derived metabolites in the host are also mentioned in the section 2.1. in the revised manuscript.
- Part 2.3.1, It is recommended to enumerate these pathways in detail of brain to gut signaling or top-down.
Response – The contents under the section 2.3.1. brain to gut signalling or top - down pathway are re-written in the revised manuscript.
- Part 3.1, It is recommended that the authors provide more detailed description of the role of gut dysbiosis in stroke.
Response – A detailed description on the role of gut dysbiosis in stroke is explained clearly with new sub-headings under the section 3.1. in the revised manuscript.
- Part 4, “Preclinical Studies on the Correlation of Gut Dysbiosis with Stroke”, A lot of clinical cases were described to prove the changes of gut microbiota in Stroke. These changes are easy to observe, can you explain these connections using further mechanisms or pathways?
Response – As per the reviewer’s comment, a separate section (3.3.) on the signaling pathways involved in gut dysbiosis associated with stroke pathology and prognosis is described in the revised manuscript.
- The description" therapeutic Strategies of probiotics in the Modulation of Intestinal Microbiota for the Prevention and Treatment of Stroke", There are many kinds of probiotics, such as Bacillus, Lactobacillus, etc. Can they all as therapeutic strategies in the modulation of intestinal microbiota for the prevention and treatment of stroke? In addition, it seems that different kinds of species, methods of using, and dosages of probiotics will lead to different clinical effects, authors should describe this strategy in detail and accurately in this section for guidance.
Response – We appreciated the reviewer’s excellent comments. Probiotics and prebiotics are reported to establish eubiosis in stroke patients and the evidence is detailed within the text. However, the precise mechanisms for these beneficial effects on regaining redox balance, reducing oxidative stress and inflammation, and regulating the neurotransmitters levels are still under study. We acknowledge that different genera and species have been used in probiotic and prebiotic formulations, as well as heterogeneity in dosage and mode of intake are mentioned in the revised manuscript in section 6.1.1. It should also be kept in mind that the clinical effects of the probiotics and prebiotics are not described clearly across all studies in a consistent manner.
Reviewer 3 Report
The article by Chidambaram et al., 2022 “The influence of gut dysbiosis in the pathogenesis and management of Ischemic Stroke” was carefully reviewed and suggested the following changes.
- There are several articles already published on this aspect, what is the difference, please justify it?
- Please avoid using long sentences.
- There are some grammar mistakes, that need to be corrected.
- Line 64, 65 and other places, words are mixed together, please make a space.
- English of the manuscript needs to be restructured.
- Please upload the plagiarism report
- Section 3.2, the authors mentioned inflammatory and immune cells, please enlist them.
- Please make the pixels of figure-1 is more prominent and use light colors.
- The inflammatory response is regulated by the Nk-KB pathway which is not mentioned in the manuscript, please highlight its function.
- Briefly, shed light on the role of natural bioactive compounds used as therapeutic agents against stroke problems.
Author Response
Comments and Responses
Title : The influence of gut dysbiosis in the pathogenesis and management of Ischemic Stroke
Reviewer -3
The article by Chidambaram et al., 2022 “The influence of gut dysbiosis in the pathogenesis and management of Ischemic Stroke” was carefully reviewed and suggested the following changes.
- There are several articles already published on this aspect, what is the difference, please justify it?
Response – We agree that this is a highly topical subject. Nonetheless, our manuscript is more comprehensive than the previous articles, providing detailed information on the pathogenetic link between stroke and gut dysbiosis. We have provided detailed molecular evidence on these links from both clinical and preclinical perspectives. We also described in depth the pathogenic mechanisms of gut dysbiosis in stroke patients from the prodromal phase to the symptomatic phase, and how gut dysbiosis affects the post-stroke prognosis. Further, we have provided up-to-date information on novel therapeutic strategies that can reverse gut dysbiosis and improve post-stroke outcome. Thus, this manuscript offers a strong blend of pathology, biochemical and pharmacological studies that provide thorough mechanistic insights.
- Please avoid using long sentences.
Response – We appreciated the comment and thus have shortened some sentences for ease of reading.
- There are some grammar mistakes, that need to be corrected.
Response – We have corrected any obvious grammatical errors.
- Line 64, 65 and other places, words are mixed together, please make a space.
Response – Mixed words are corrected in the revised manuscript. The lines 64 and 65 are re-written addressing the comments of reviewer 1 as well.
- English of the manuscript needs to be restructured.
Response – We have re-structured some of the text to improve comprehension.
- Please upload the plagiarism report
Response – As per the reviewer comment, plagiarism report generated using Turnitin software is uploaded along with the revised manuscript.
- Section 3.2, the authors mentioned inflammatory and immune cells, please enlist them.
Response – As per reviewer suggestions, the list of inflammatory and immune cells is added in the revised manuscript under section 3.2.
- Please make the pixels of figure-1 is more prominent and use light colors.
Response – Figure.1 is redrawn with light colors and improved pixels.
- The inflammatory response is regulated by the Nk-KB pathway which is not mentioned in the manuscript, please highlight its function.
Response – In the revised manuscript under section 3.3., the importance of nuclear factor (NF)-κB signaling pathway and other pathways associated with stroke and gut dysbiosis are mentioned.
- Briefly, shed light on the role of natural bioactive compounds used as therapeutic agents against stroke problems.
Response – Both preclinical and clinical studies addressing the neuroprotective effects of natural bioactive compounds against stroke are added under section 6.3. in the revised manuscript.
Round 2
Reviewer 2 Report
It could be accepted in the present version.
Reviewer 3 Report
The article was revised as per suggestion. Now, it can be accepted.